# Comparative Corrosion Characterization of Hybrid Zinc Coatings in Cl⁻-Containing Medium and Artificial Sea Water

Nelly Boshkova [1], Kamelia Kamburova [1], Tsetska Radeva [1], Silviya Simeonova [2], Nikolay Grozev [2], Maria Shipochka [3] and Nikolai Boshkov [1,*]

[1] Institute of Physical Chemistry "R. Kaishev", Bulgarian Academy of Sciences (BAS), 1113 Sofia, Bulgaria

[2] Faculty of Chemistry and Pharmacy, Sofia University, 1164 Sofia, Bulgaria

[3] Institute of General and Inorganic Chemistry, Bulgarian Academy of Sciences, "Acad. G. Bonchev" St. Bl. 11, 1113 Sofia, Bulgaria

* Correspondence: nboshkov@ipc.bas.bg

**Abstract:** The presented investigations demonstrate the corrosion behavior and protective ability of hybrid zinc coatings specially designed for combined protection of low-carbon steel from localized corrosion and biofouling. Polymer-modified copper oxide (CuO) nanoparticles as widely used classic biocide are applied for this purpose, being simultaneously electrodeposited with zinc from electrolytic bath. The corrosion behavior of the hybrid coatings is evaluated in a model corrosive medium of 5% NaCl solution and in artificial sea water (ASW). Scanning electron microscopy (SEM) and atomic force microscopy (AFM) are used to characterize the surface morphology of pure and hybrid zinc coatings. Contact angle measurements are realized with an aim to determine the hydrophobicity of the surface. X-ray photoelectron spectroscopy (XPS) is applied for evaluation of the chemical composition of the surface products appearing as a result of the corrosion treatment. Potentiodynamic polarization (PDP) curves and polarization resistance (Rp) measurements are used to estimate the protective characteristics in both model corrosive media. The results obtained for the hybrid coatings are compared with the corrosion characteristics of ordinary zinc coating with the same thickness. It was found that the hybrid coating improves the anticorrosion behavior of low-carbon steel during the time interval of 35 days and at conditions of external polarization. The tests demonstrate much larger corrosion resistance of the hybrid coating in ASW compared to 5% NaCl solution. The obtained results indicated that the proposed hybrid zinc coating has a potential for antifouling application in marine environment.

**Keywords:** hybrid zinc coating; corrosion; zinc; CuO nanoparticles; 5% NaCl solution; artificial sea water

## 1. Introduction

Biofouling and corrosion are undesirable processes that lead to microbiologically and chemically induced degradation of metal structures in marine environment. Incorporation of biocides into antifouling coatings have a positive long-term effect on metal protection due to the prolonged release of the biocide. Some of the most studied inorganic biocides are copper and copper-containing materials due to the well-known cooper toxicity to marine microorganisms [1–3]. In the seawater, the antifouling performance of the copper-based coatings is realized mainly through the release of cuprous and copper ions reacting with Cl⁻ ions [4].

Electrodeposition of protective zinc and hybrid zinc coatings is a common way to minimize the degradation of steel structures in coastal and marine zones [5,6]. Incorporation of copper oxide (CuO) nanoparticles into zinc coatings electrodeposited on steel seems to be a promising way to ensure controlled release of copper ions in a case of corrosion attack. CuO nanoparticles may dissociate into $Cu^{2+}$ ions, which significantly affect the growth of aquatic

microorganisms at higher concentrations [7]. Earlier studies reported that copper oxide also influenced the corrosion protection of copper, iron, zinc, and other metals by hindering the penetration of aggressive corrosive agents deeply inside the metal structures [8]. For example, the incorporation of CuO nanoparticles into the matrix of a protective zinc coating has recently been shown to remarkably increase the anticorrosion efficacy of the coating for mild steel in a Cl-containing environment (3.65 wt.% NaCl solution) [9].

Electrolytic cathodic deposition of CuO nanoparticles in the zinc matrix requires stabilization of the particles suspension against aggregation since smaller particles were found to provide better protective ability of the coatings than the agglomerated ones [10]. The suspensions stability depends mainly on the electrostatic (and/or steric) repulsion between the likely charged particles. A polyelectrolyte adsorption can aid dispersion by increasing positive charge on the particle surface, which can be also useful for successful electrodeposition on the metal surfaces. For example, adsorption of polyethylenimine (PEI) on the CuO particles can improve their stability against agglomeration by increasing the particles surface charge density. Working as a charging, dispersing, and film-forming agent, PEI has also been found to improve the protection from corrosion of steel alone or as a part of hybrid coatings containing incorporated metal oxide particles [11,12].

The choice of PEI was also based on the fact that it is a good corrosion inhibitor for steel in near neutral chloride media [11]. Its inhibition effectiveness at a concentration of $10^{-3}$ g/L reach 90% after 1 month of immersion in a 3% NaCl solution. The presumption is that the polymer forms a dense layer on the steel surface, which prevents in such a way the penetration of the aggressive agents deeply inside. Antifouling effect of zinc has also been reported due to antimicrobial and antibacterial properties of zinc-based materials despite the generally lower toxicity of zinc to aquatic organisms compared to copper [13,14].

The aim of the present work is to demonstrate and compare the corrosion resistance and protective ability of ordinary and of a newly developed hybrid zinc coating (containing incorporated polymer-modified CuO nanoparticles in the zinc matrix) on low-carbon steel in two selected model media—5% NaCl (generally causes the appearance of localized corrosion) and artificial sea water (ASW—for checking the corrosion process at conditions suitable for emergence and development of biofouling).

## 2. Materials and Methods

### 2.1. Materials and Preparation of Stable CuO Suspension

CuO nanopowder (<50 nm particle size) and poly(ethylenimine) (PEI, Mw = 25 kDa) were purchased from Sigma-Aldrich (Darmstadt, Germany). The PEI used in these experiments is a branched polyamine, composed of primary, secondary, and tertiary amines that are protonated in neutral and acidic conditions. Stable CuO suspension was prepared by addition of CuO nanopowder (1 g/L) to a PEI aqueous solution of concentration 1 g/L, and the suspension was ultrasonicated in ice for 15 min aiming to improve the particles dispersion.

### 2.2. Characterization of CuO Nanoparticles Dispersion

The working concentration of the CuO suspensions with and without PEI was set to 0.1 g/L, and pH was adjusted to 7.5 to minimize the effects of CuO dissolution and particles agglomeration. The procedure for characterization is described in more detail in [15]. The size determination of CuO nanoparticles in the aqueous suspension was realized by transmission electron microscopy (HR STEM JEOL JEM 2100, Tokyo, Japan). Two other important parameters—hydrodynamic diameter and zeta potential of the nanoparticles—were evaluated by dynamic light scattering (DLS) and laser Doppler velocimetry (Zetasizer Pro Red Label, Malvern Panalytical Ltd., Malvern, United Kingdom). As a light source, a HeNe laser was applied, and intensity was measured by a detector at 173°.

### 2.3. Electrodeposition of Hybrid Zinc Coatings on Steel

The hybrid and ordinary (for comparison) zinc coatings were obtained on a low-carbon steel sample with a working area of 6 cm$^2$ (sizes $3 \times 1 \times 0.1$ cm). The starting and slightly acidic

zinc electrolyte has a composition (g/L): $ZnSO_4 \cdot 7H_2O$—150; $(NH_4)_2SO_4{}^-$—30; $H_3BO_3$—30. Two additives, AZ1 (wetting agent) and AZ2 (brightener), were also used. The electrodeposition process was realized in a glass cell (300 mL volume) at the following conditions: pH 4.5–5.0, cathodic current density of 2 $A/dm^2$, no stirring, metallurgic zinc anodes, and room temperature. In order to obtain hybrid coating, a solution containing $10^{-1}$ g/L polymer-modified CuO nanoparticles (coated with PEI) was added to the starting electrolyte. Final thickness of both coating types was ~12 μm.

### 2.4. Surface Morphology

The surface morphology of the ordinary and hybrid zinc coatings before and after corrosive treatment was evaluated with scanning electron microscopy (Oxford Instruments, Oxford, UK) by using of INCA Energy 350 unit.

### 2.5. Corrosion Characterization and CVA Studies

The corrosion characterization of the investigated coatings was realized with well-known electrochemical methods: potentiodynamic (PDP) polarization curves and polarization resistance (Rp) measurements. The results obtained were compared with that of the ordinary zinc. The tests were carried out with computerized PAR unit "VersaStat 4". Saturated calomel electrode (SCE) was the reference, and platinum plate was the counter electrode. The scan rate in cathodic and anodic direction was 1 mV/s. The curves were stopped after visual appearance of the steel substrate, checking by "naked eye".

Polarization resistance (Rp) is used for evaluation of the protective ability of the coatings. Its value is inversely proportional to the corrosion current density. Higher Rp values are a sign for better protective ability and lower corrosion rate (lower corrosion current density) as well. The Rp values were measured during a period of 35 days.

Cyclic voltammetry (CVA) tests were carried out in the potential interval between −2 and 0 V with a scan rate of 10 mV/s.

### 2.6. XPS Measurements

The film composition and electronic structure were investigated by X-ray photoelectron spectroscopy (XPS). The measurements were carried out on AXIS Supra electron-spectrometer (Kratos Analitycal Ltd., Manchester, UK) using monochromatic AlKα radiation with a photon energy of 1486.6 eV and charge neutralization system. The binding energies (BE) were determined with an accuracy of ±0.1 eV. The chemical composition in the depth of the films were determined monitoring the areas and binding energies of Zn2p, O1s, Cu2p, Cl2p, and C1s photoelectron peaks. Using the commercial data-processing software of Kratos Analytical Ltd., the concentrations of the different chemical elements (in atomic %) were calculated by normalizing the areas of the photoelectron peaks to their relative sensitivity factors. More information about the principles and features of the methods is described elsewhere [16,17].

### 2.7. Atomic Force Microscopy (AFM) Investigations

AFM imaging was performed on the NanoScope V system (Bruker Ltd., Bremen, Germany) operating in tapping mode in air at room temperature. Silicon cantilevers (Tap 300Al-G, Budget Sensors, Innovative solutions Ltd., Sofia, Bulgaria) with 30 nm thick aluminum reflex coatings were used. According to the producer's datasheet, the cantilever force constant and the resonance frequency are in the range 40 N/m and 300 kHz, respectively. The tip radius was less than 10 nm. The scan rate was set at 1 Hz, and the images were captured in height mode with 512 × 512 pixels in JPEG format. Subsequently, all images were flattened by using NanoScope software (Bruker Inc., Birrica, MA, USA). The same software was also used for section and roughness analysis.

### 2.8. Contact Angle Measurements

Measurements of the contact angle of small water drops (volume ≈ 3 microliters) were made using an automatic goniometer/tensiometer (Model 290, Ramè—Hart Ltd., Succasunna, NJ, USA) with DROP images Advanced v. software 2.4 (Succasunna, NJ, USA) at room temperature. The contact angles of 10 consecutive drops of 3 µL positioned at random locations of the samples were measured. A mean angle and a mean error were taken from them.

### 2.9. Test Media and Reproducibility

Electrochemical tests were carried out in two model corrosive media—5% NaCl solution (pH value of ~6.7) and artificial sea water (ASW)—with a content according to ASTM D 665. The experimental results are an average from the data of five samples per type, i.e., either zinc or hybrid zinc coatings.

## 3. Results and Discussion

### 3.1. Characterization of CuO Nanoparticles Dispersion

For the particles size and charge measurements, the suspension was diluted to concentration of 0.1 g/L, and the pH was set to 7.5 to minimize the CuO dissolution, showing a significant increase below pH~7 [14]. According to TEM analysis (Figure 1), the CuO nanoparticles are nearly spherical and formed aggregates in water. The mean diameter of a separate particle is about $65 \pm 13$ nm, while the diameter of most aggregates in water is much larger—approximately $225 \pm 4$ nm according to the DLS measurement. It is well-known that the hydrodynamic size of the nanoparticles is larger than the size of the primary dry particles due to their interaction with surrounding solution [18,19]. The CuO nanoparticles have a zeta potential value of about +30 mV, which is close to the results of previous investigations on the same CuO nanoparticles [20,21]. It is known that the potential values of $\pm 30$ mV are accepted as minimum zeta potentials for obtaining stable suspensions due to strong electrostatic repulsion between likely charged particles.

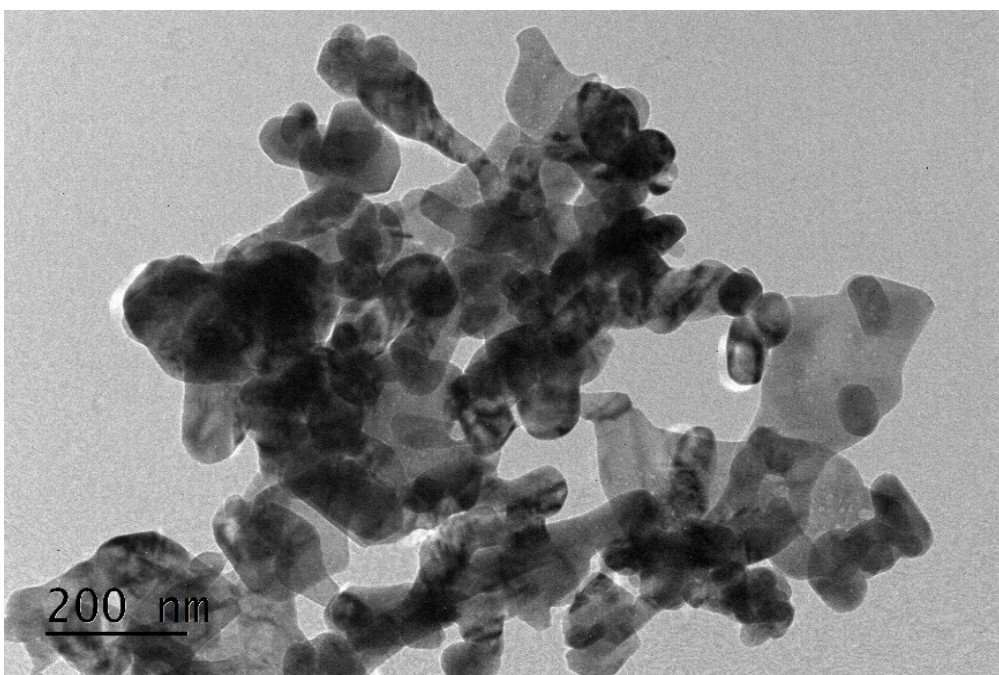

**Figure 1.** TEM image of CuO nanoparticles in water (0.1 g/L, pH 7.5).

Keeping in mind that the surface charge of the particles might decrease in the bath solution (due to the increase in ionic strength), PEI was added to the CuO suspension to

minimize further particles agglomeration. The zeta potential of the CuO nanoparticles increases from +30 mV to about +55 mV due the adsorption of PEI chains. Figure 2 shows slight increase in the hydrodynamic diameters of the CuO particles from 225 ± 4 to 240 ± 5 nm, which confirms the formation of thin (~7–8 nm) adsorption of PEI layer on the particle surface and stabilization of the suspension against further agglomeration. The mechanism of adsorption of positively charged PEI on the likely charged CuO nanoparticles has already been discussed in a work focused on electrophoretic deposition of CuO particles on metallized (and steel) surfaces using PEI as a dispersing agent [22]. According to the authors, dative bonds are formed between the $Cu^{2+}$ ions on the CuO nanoparticle surface and the un-protonated amine groups of PEI resulting in surface complexation of CuO. The protonated amino groups are responsible for stabilization of the suspension against agglomeration.

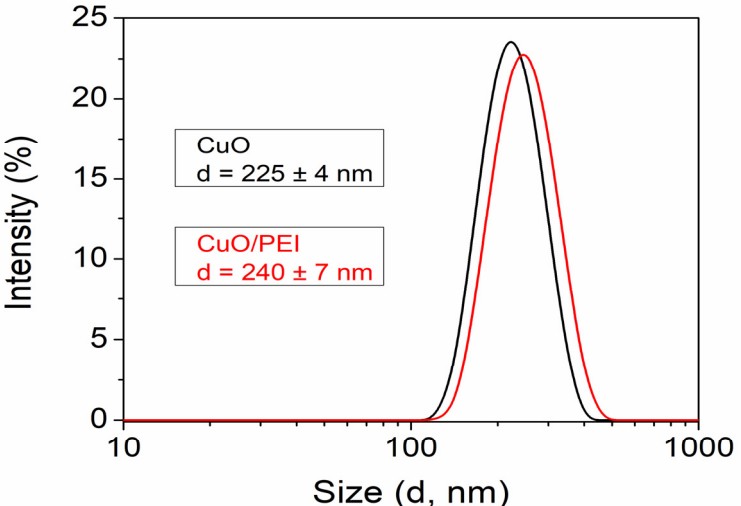

**Figure 2.** Particle size distribution.

### 3.2. Surface Morphology

The surface morphology of the hybrid and ordinary zinc coatings is presented in Figure 3. Generally, both surfaces look relatively similar. However, it is obvious that some zones with embedded white spheres appear on the surface of the hybrid zinc sample. This can be expected due to the presence of positively charged (polymer modified) CuO nanoparticles in the starting electrolyte, which deposit simultaneously with the zinc ions during the electrodeposition.

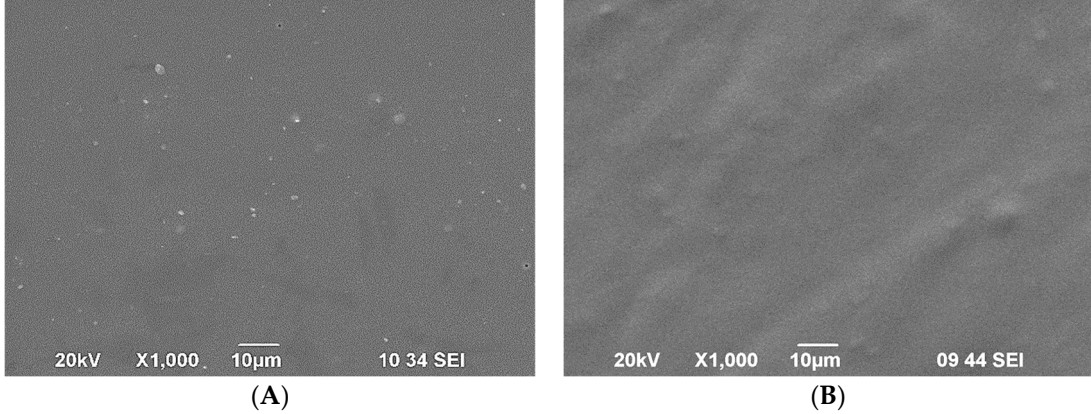

**Figure 3.** SEM micrographs of (**A**) hybrid zinc and (**B**) ordinary zinc coatings.

### 3.3. Cyclic Voltammetry (CVA) Studies

The results obtained with CVA are reported in Figure 4A. The experiments are carried out in the starting electrolytes (suspensions) for electrodeposition of ordinary and hybrid zinc coatings, respectively. It can be seen that the electrodeposition of the ordinary zinc occurs at higher current density and is significantly hampered (overpolarization), i.e., starts at more negative potential values compared to the hybrid coating. The cathodic deposition process of the hybrid coating starts at more positive potential values (depolarization) and is realized at much lower cathodic current density.

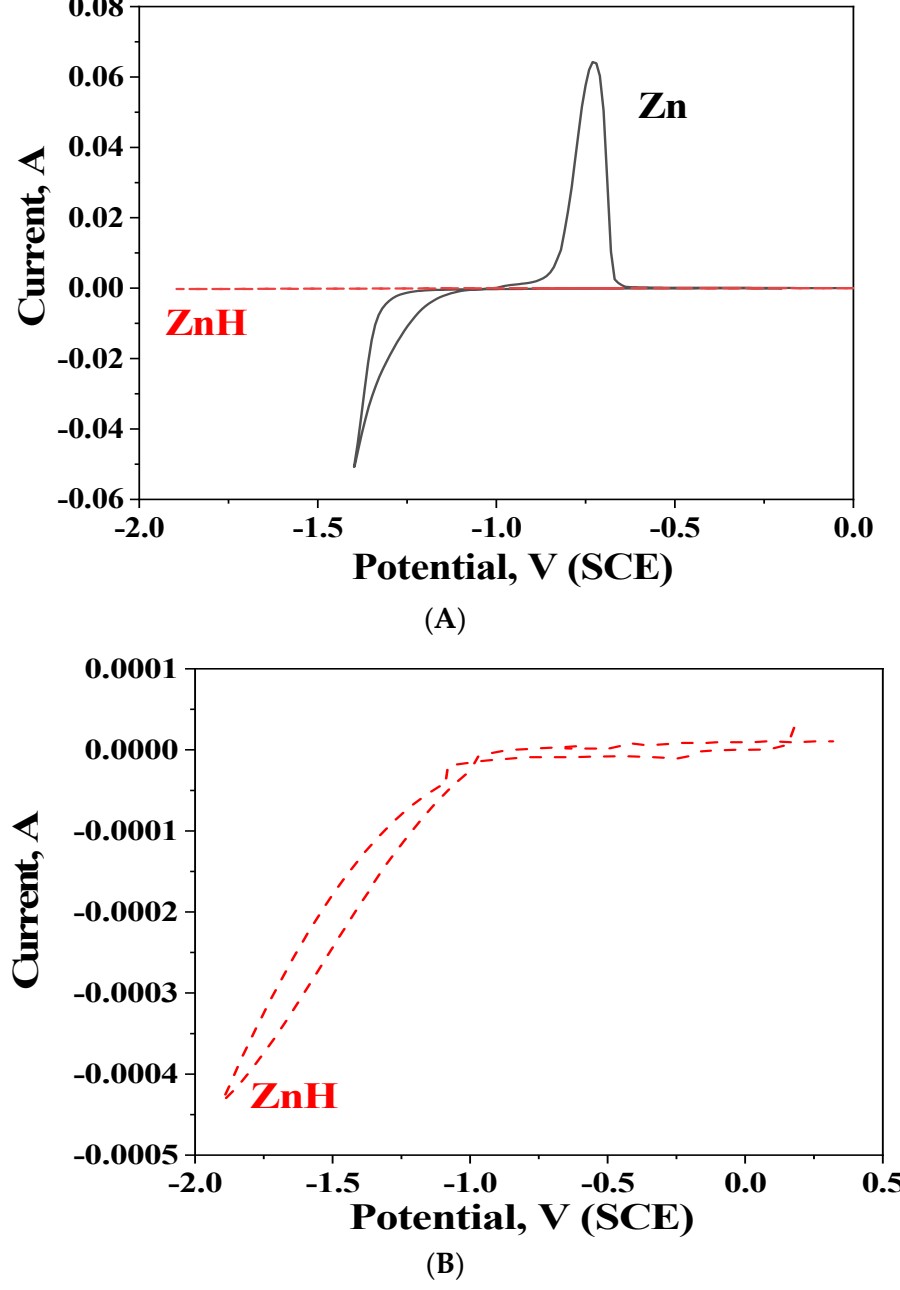

**Figure 4.** (**A**) Cyclic voltammetry of hybrid (ZnH) and ordinary zinc coatings. (**B**) Cyclic voltammetry of hybrid (ZnH).

The anodic branch of the same CVA curve confirms the presence of thin zinc hybrid coating due to weakly expressed dissolution process—Figure 4B. One probable reason for such behavior is the presence of CuO nanoparticles, the potential of which is more positive

compared to the zinc ions, so they electrodeposit first on the steel substrate. Thereafter (since the nanoparticles are greater in size compared to the zinc ions), the steel surface will be partially blocked, resulting in a slower deposition rate of the hybrid coating.

### 3.4. Potentiodynamic Polarization (PDP) Curves and Polarization Resistance (Rp) Measurements

Potentiodynamic polarization curves of both coating types in the selected corrosive media are demonstrated in Figure 5. In 5% NaCl solution, the potential of the ordinary zinc coating is placed at more positive values, and its corrosion current is lower compared to the case when this sample is immersed in ASW (see the most important electrochemical parameters in Table 1). It is obvious that significant difference appear in the anodic curves of the zinc and hybrid zinc coatings in both corrosive media: the anodic curve of the ordinary zinc coating in 5% NaCl solution passes through a maximum at a potential of about −0.8 V, and thereafter, the coating dissolves until exposing the steel substrate at about −0.62 V. Contrary to this, the anodic current density of this coating in ASW increases gradually until the complete dissolution in the potential area, about −0.4 V; i.e., the coating seems to be more resistive in that medium at these conditions. A similar effect was reported in [23–25]. The reason for this observation is most probably the composition of the corrosive products, which appear on the samples surface during the test. In the case of 5% NaCl solution, the most probable corrosion product is zinc hydroxide chloride (ZHC)—$Zn_5(OH)_8Cl_2 \cdot H_2O$— well-known from other investigations [26,27]. This compound has a very low product of solubility and ensures better protection of the substrate impeding the penetration of the aggressive chloride ions deeply inside. In the ASW, due to the presence of other ions such as $Ca^{2+}$, $Mg^{2+}$, $Sr^{2+}$, and $K^+$ as well as $HCO_3^-$, $SO_4^{2-}$, $Br^-$, and $F^-$, different corrosion products will appear, leading to greater surface and composition inhomogeneity.

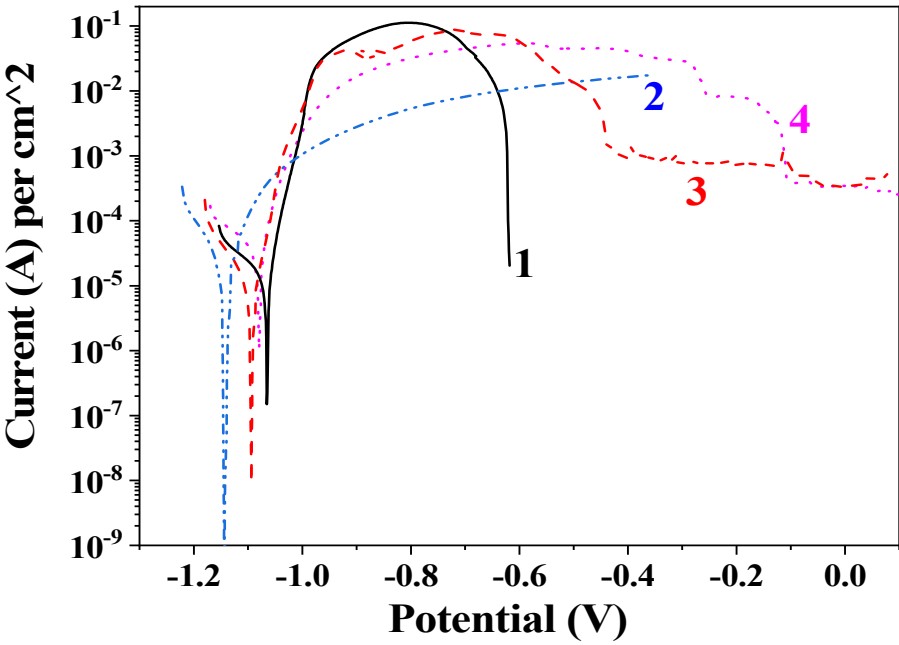

**Figure 5.** Potentiodynamic polarization curves of ordinary zinc coating in (**1**) NaCl and (**2**) ASW and hybrid zinc coating in (**3**) NaCl and (**4**) ASW.

The hybrid zinc coating also demonstrates some peculiarities in both corrosive media: in 5% NaCl solution, the anodic current density curve passes through a maximum at about −0.7 V, and thereafter, a passive zone occurs in the interval between −0.44 and 0.1 V. In the ASW, this coating dissolves (accelerated anodic dissolution) for a longer period (up to −0.1V) and thereafter passivates. The corrosion current density for the hybrid coating is lower in 5% NaCl compared to ASW.

**Table 1.** Electrochemical parameters obtained from potentiodynamic polarization measurements.

| No. | Sample/Medium | $I_{corr}$, $A \cdot cm^{-2}$ | $E_{corr}$, V | $I_{pass}$, $A \cdot cm^{-2}$ |
|-----|---------------|------------------|-----------|------------------|
| 1 | Zn/5% NaCl | $1.8 \times 10^{-5}$ | −1.07 | - |
| 2 | Zn/ASW | $2.1 \times 10^{-5}$ | −1.14 | - |
| 3 | ZnH/5% NaCl | $9.5 \times 10^{-6}$ | −1.09 | $7.2 \times 10^{-4}$ |
| 4 | ZnH/ASW | $3.2 \times 10^{-5}$ | −1.08 | $3.2 \times 10^{-4}$ |

Generally, it can be concluded that the presence of polymer modified CuO nanoparticles positively influences the corrosion characteristics of the zinc coating on steel, leading to the appearance of a passive zones and longer anodic curves at conditions of external anodic polarization.

Polarization resistance data obtained after 35 days' immersion of the samples in both corrosive media are shown in Figure 6. It is obvious that ordinary zinc coating demonstrates close Rp values in 5% NaCl and ASW solutions during the whole period of investigation. Only at the beginning of the test (1–15 days), the polarization resistance in ASW is higher and reaches ~1000 $\Omega \cdot cm^2$. Thereafter, this parameter was held back at about 750 $\Omega \cdot cm^2$ and at the end of the test is ~ 850 $\Omega \cdot cm^2$ in both media.

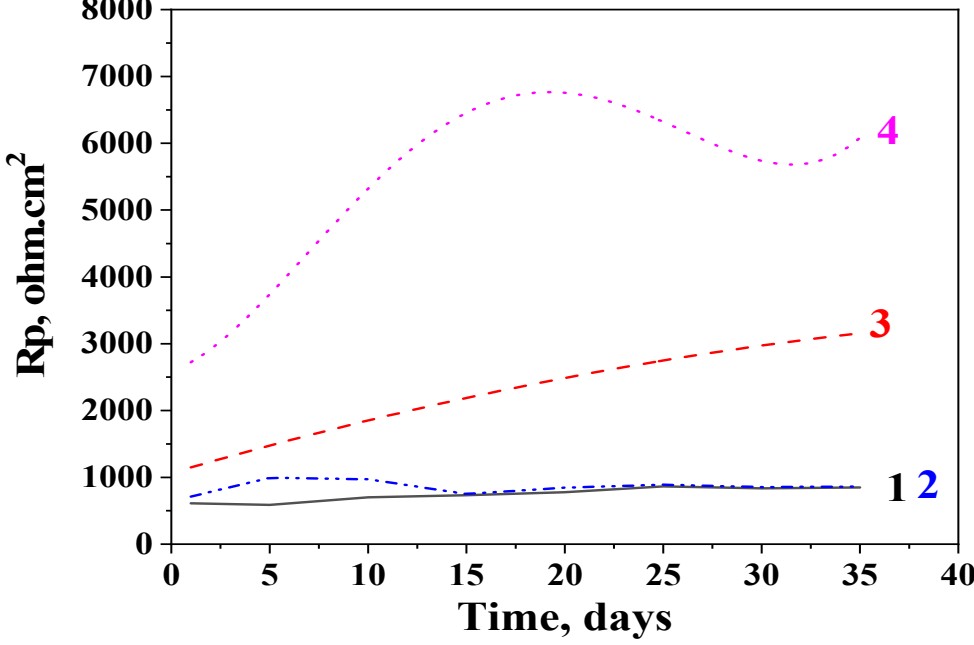

**Figure 6.** Polarization resistance of ordinary zinc coating in (**1**) NaCl and (**2**) ASW and of hybrid zinc coating in (**3**) NaCl and (**4**) ASW.

The experimental results of the hybrid zinc samples demonstrate improved corrosion resistance compared to the ordinary zinc coating, which is clearly expressed in the ASW solution. In the latter, the Rp values pass through a maximum after 20 days' immersion (Rp ~ 7000 $\Omega \cdot cm^2$), followed by a decreasing tendency (till 33-th day) and then slight increase at the end of the test (~6000 $\Omega \cdot cm^2$). In 5% NaCl solution, the polarization resistance increases during the whole immersion period, reaching 3000 $\Omega \cdot cm^2$ at the final stage of the experiment.

These results could be explained with the appearance of corrosion products in the corrosive media, as already discussed above. However, the difference is that, in the case of the immersion test ("open-circuit" conditions), the forming and appearance of these compounds occurs slowly, contrary to the case of external polarization.

SEM images of hybrid zinc coatings after corrosive treatment in the model test media are demonstrated in Figure 7. In the case of 5% NaCl, the surface seems relatively even and almost fully covered by corrosion products, which slows down the penetration of the chloride ions toward the steel substrate. It could be supposed that, as a result of the corrosive treatment, the coating is partially transformed into a mixed layer simultaneously containing a corrosive product $Zn_5(OH)_8Cl_2 \cdot H_2O$ and polymeric materials retained from the encapsulation process of the CuO nanoparticles. In the case of ASW, parts of the coating are still not transformed in corrosive products although the coating is seriously damaged/covered with holes and cracks. Corrosion products appear in some places on the surface but do not cover the whole sample area. This result partially explains the changes in the course of the hybrid coating behavior in ASW compared to the coating in 5% NaCl, where the change in the Rp values is steadier. It can be concluded that the composition of the newly formed corrosion products differs in both media, leading to some peculiarities in the corrosion behavior.

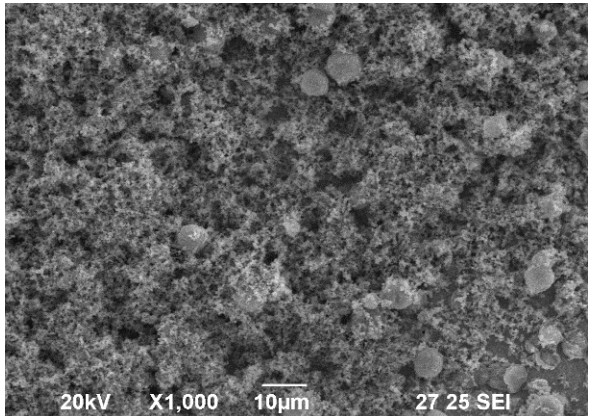
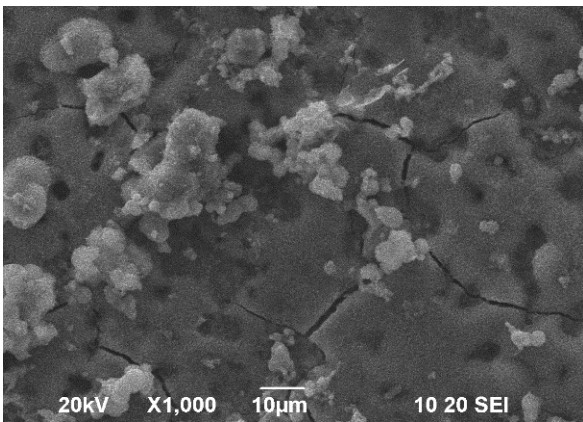

Hybrid zinc coating/5% NaCl        Hybrid zinc coating/ASW

**Figure 7.** SEM images of hybrid zinc coatings after immersion in the corrosive media.

Different corrosion resistance of hybrid aluminum oxide/titanium nitride coatings produced on stainless steel has earlier been registered in 0.5 M $H_2SO_4$, 1 M HCl, and 0.75 M NaCl solutions [28]. The results showed an improvement of the corrosion resistance of the hybrid coatings (compared to pure steel) in most cases except in NaCl solutions. According to [28], the adhesion of the coating was considerably affected by the NaCl solution, and initiation of crevice and pitting corrosion was observed.

*3.5. Atomic Force Microscopy (AFM) and Contact Angle Investigations*

The AFM images of the ordinary and hybrid zinc coatings were presented in Figure 8. The morphology of the zinc sample with a scan area of $10 \times 10$ μm$^2$ (Figure 8 left) is rough (with the presence of spherical structures) compared to the morphology of the hybrid coating (Figure 8 right). In the presence of CuO nanoparticles, the morphology of the hybrid coating changes: the observed spherical structures in the zinc coating transformed into denser zones. It can be seen that the surface of the hybrid coating is dense, with the presence of high and low areas.

For a more precise comparison of the surface morphology, the two types of coatings were scanned with a scanning area of $5 \times 5$ μm$^2$ (Figure 9). It is clearly observed again that the surface of the zinc sample has a "granular" structure, with many grains of same size. In the presence of CuO nanoparticles, the morphology of the zinc matrix changes, with the observed smaller structures clustering into larger regions.

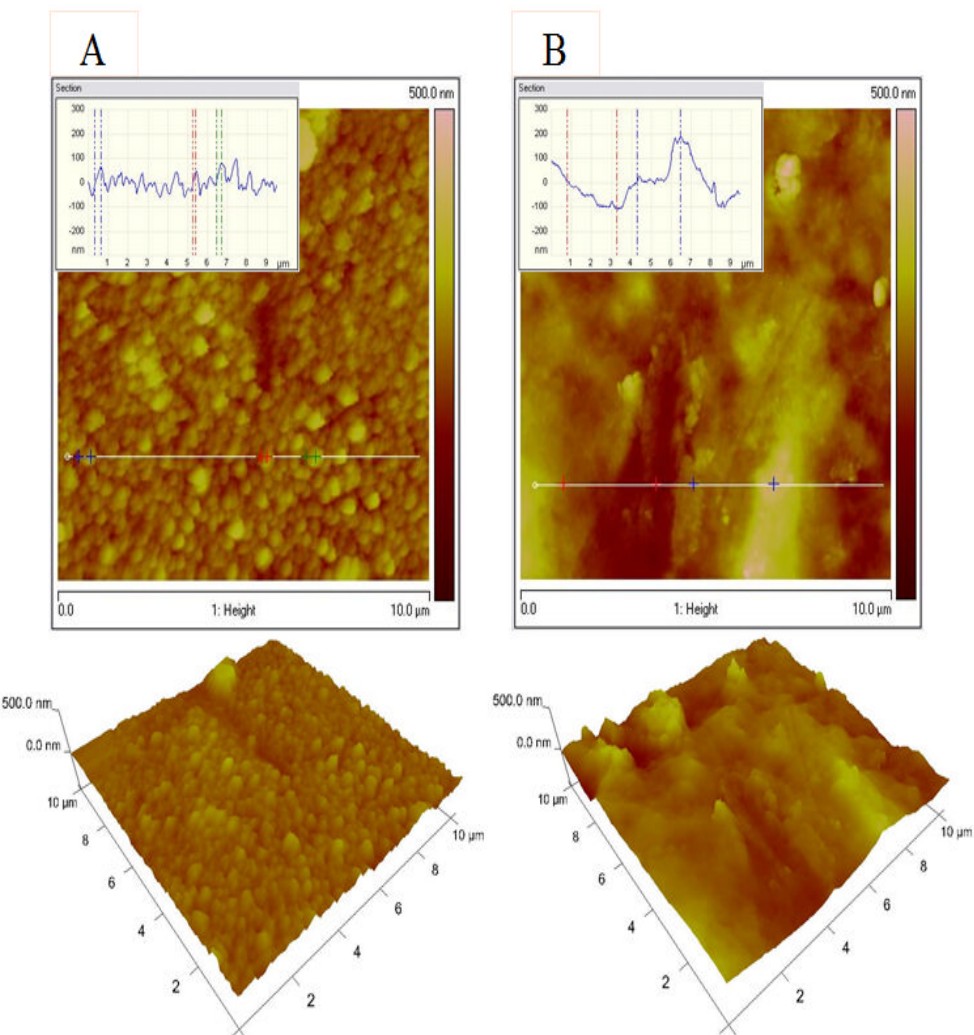

**Figure 8.** AFM topography of ordinary zinc (**A**) and hybrid zinc (**B**) coatings: 2D images, 3D images, and section analysis.

Photographs of water droplets on the surface of the ordinary and hybrid zinc coatings were compared by measuring the contact angles (Figure 9). One could note that the hybrid coating with CuO nanoparticles incorporated into the zinc matrix changes the contact angle from 94° for the ordinary zinc sample to 138° for the hybrid coating. Most probably, the more hydrophobic surface profile of the hybrid coating is due to the different topography compared to the ordinary zinc coating.

*3.6. XPS Studies*

The surface compositions and chemical states of the samples were investigated by XPS method after 35 days' immersion of the coatings in both corrosive media. The spectra of Zn, O, Cu, Cl, C, etc., are registered on the surface. The photoelectron spectra of Zn2p presents two peaks with binding energies at ~1022.0 eV for $Zn2p_{3/2}$ and ~1045.0 eV for $Zn2p_{1/2}$. Observed peaks positions and spin orbital splitting between $2p_{3/2}$ and $2p_{1/2}$ of ~23.0 eV are characteristics of ZnO. In the spectra of ZnO, at higher binding energies, a shoulder is observed, which is due to the bond of zinc with hydroxyl groups at 1023 eV and probably also with chlorine in the form of zinc hydroxide chloride (ZHC)—$Zn_5(OH)_8Cl_2 \cdot H_2O$ at 1024 eV—well-known from other investigations (Figure 10) [26,27].

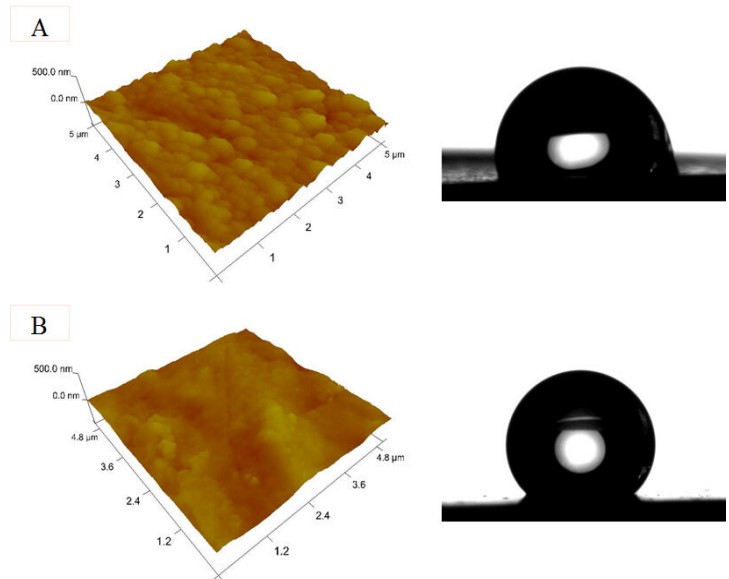

**Figure 9.** AFM 3D images and photographs of water drops on zinc (**A**) and hybrid zinc (**B**) coatings.

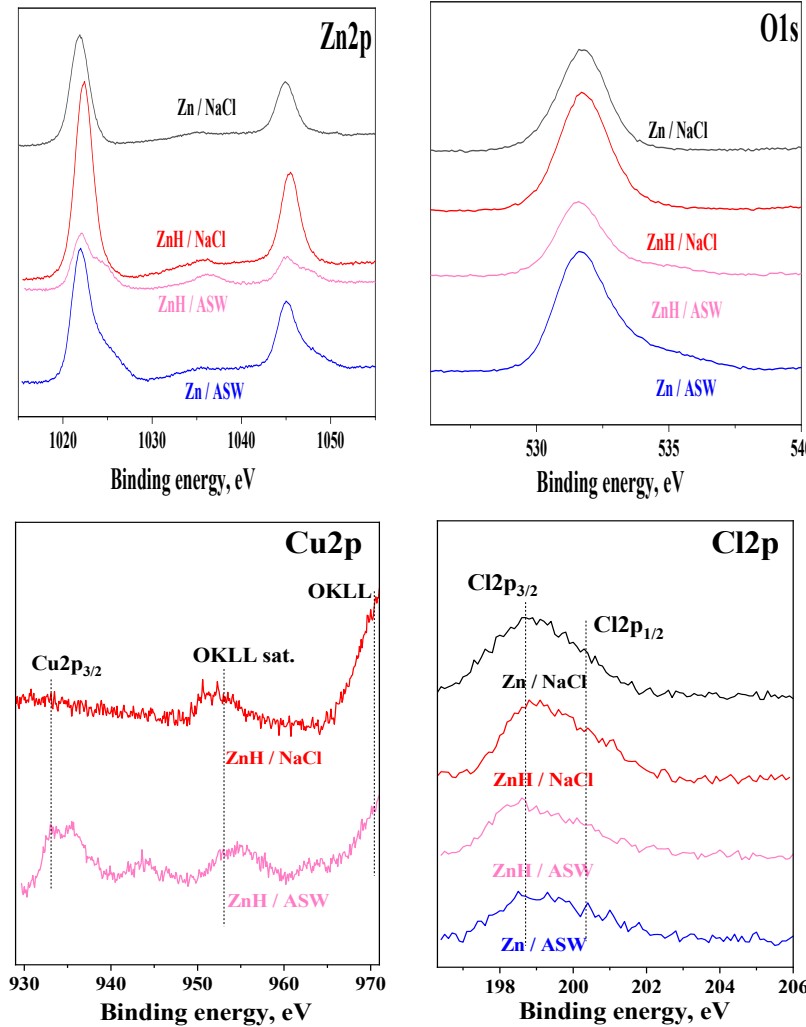

**Figure 10.** XPS spectra of the corrosive treated Zn and ZnH samples.

The O1s peaks are wide and asymmetric and can be fitted into several components, which are oxygen in the metal lattice of ZnO (~531.4 eV), oxygen bonded with hydroxyl group in $Zn(OH)_2$ (~532.3 eV), carbonates (~533.8 eV), and oxygen in the water molecules (~535.7 eV). There is one more component of O1s spectra for the composite sample treated in artificial sea water, which is attributed to CuO (933.2 eV) and probably to $Cu(OH)_2$ (935.3 eV) (Figure 10). The maximum of the Cl2p peaks has a binding energy at 198.7 eV, which is assigned to the Zn-Cl bond and to NaCl; i.e., it could be supposed that ZHC and NaCl simultaneously appear in the sample, while NaCl is left from the corrosive treatment. The reason for the absence of CuO in the case of 5% NaCl solution could be the more aggressive nature of this medium, which can be also confirmed by the SEM investigations, i.e., ZnH coating is practically transformed in ZHC layer. In Figure 10, showing the copper spectrum, the Auger oxygen line OKLL and its satellite are visible at higher binding energies.

## 4. Conclusions

Optimum conditions were found for preparation of a stable suspension of CuO nanoparticles aimed to be incorporated into ordinary zinc coating. Due to the well-expressed anti-bactericide effect of CuO and the sacrificial nature of the zinc, the combined anticorrosion and biofouling protection of low-carbon steel could be proposed. Hybrid coating was electrodeposited from a slightly acidic electrolyte by simultaneous electrode-position of polymer-modified CuO nanoparticles and zinc.

The hybrid zinc coating was found to improve the anticorrosion behavior of low-carbon steel in two well-known and applied corrosive media—neutral 5% NaCl solution (causes generally localized corrosion) and artificial sea water (for checking the corrosion process at conditions suitable for emergence and development of biofouling) for a time interval of 35 days at open-circuit conditions and at external polarization. The protective action of the hybrid coating is a result most probably of the appearance of newly formed corrosion products, which are consequence of the interaction of the zinc matrix with the surrounding environment.

Barrier effects realized by the presence of these corrosion products, and the availability of polymer materials/layers in them (appearance of "mixed" layer) seems to contribute to the enhanced protective properties of the hybrid coating in both test corrosive media compared to the ordinary zinc coating. The more hydrophobic nature of the hybrid coating additionally supports the corrosion resistance in both corrosive media.

The protective characteristics of the hybrid coating are better demonstrated in ASW compared to 5% NaCl solution, most probably due to the higher aggressiveness of the NaCl environment. The obtained results indicated that the proposed hybrid zinc coating could be used for antifouling protection of low-carbon steels in a marine environment.

**Author Contributions:** Conceptualization, N.B. (Nikolai Boshkov) and T.R.; methodology, N.B. (Nelly Boshkova), K.K., S.S., N.G. and M.S.; software, S.S., N.G. and M.S.; validation, N.B. (Nelly Boshkova), K.K., T.R. and N.B. (Nikolai Boshkov); formal analysis, S.S., N.G. and M.S.; investigation, N.B. (Nelly Boshkova), K.K., N.B. (Nikolai Boshkov) and T.R.; resources, N.B. (Nikolai Boshkov); data curation, N.B. (Nikolai Boshkov); writing—T.R. and N.B. (Nikolai Boshkov); writing—review and editing, T.R. and N.B. (Nikolai Boshkov); supervision, T.R. and N.B. (Nikolai Boshkov); project administration, N.B. (Nelly Boshkova). All authors have read and agreed to the published version of the manuscript.

**Funding:** The authors express their gratitude to the project with the Fund "Scientific Investigations", Bulgaria, KP-06-China/4 (КП-06-Китай/4), "Developing novel composite materials and their surface coatings for long-term anti-corrosion/biofouling applications" for the financial support and for the possibility to publish the obtained results.

**Institutional Review Board Statement:** Not applicable.

**Informed Consent Statement:** Not applicable.

**Data Availability Statement:** Not applicable.

**Acknowledgments:** The authors express their gratitude to the project with the Fund "Scientific Investigations", Bulgaria, KP-06-China/4 (КП-06-Китай/4), "Developing novel composite materials and their surface coatings for long-term anti-corrosion/biofouling applications" for the financial support and for the possibility to publish the obtained results. The support of the European Regional Development Fund within the OP Science and Education for Smart Growth 2014–2020, Project CoE: National Centre for Mechatronics and Clean Technologies, No. BG05M2OP001-1.001-0008, is also acknowledged. Research equipment of the Distributed Research Infrastructure INFRAMAT, part of the Bulgarian National Roadmap for Research Infrastructures, supported by the Bulgarian Ministry of Education and Science, was used in these investigations.

**Conflicts of Interest:** The authors declare no conflict of interest.

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
