# Peer review of "Comparative Corrosion Characterization of Hybrid Zinc Coatings in Cl-Containing Medium and Artificial Sea Water"

_coatings, doi:10.3390/coatings12121798_

Round 1

Reviewer 1 Report

The suggested manuscript can be accepted for publication after the corrections listed below.

1. Please, provide error bars here. From my point of view, the curves look almost equal to each other. Please, pay attention to it and discuss the reason for this very slight difference.

The comment is taken into consideration and the needed text is added:

 “The mean diameter of a separate particle is about 65 ± 13 nm, while the diameter of most aggregates in water is much larger - approximately 225 ± 4 nm according to the DLS measurement. It is well known that the hydrodynamic size of the nanoparticles is larger than the size of the primary dry particles due to their interaction with surrounding solution [13,14].”

The curves in Fig. 2 look very close to each other because of the formation of thin PEI layers (7-8 nm) on the particles (aggregates) surface, which is a reasonable value for the hydrodynamic thickness of a polyelectrolyte layer adsorbed on oppositely charged surface from water solution.

The above presented explanation and evaluated thickness of the adsorbed PEI layers were added in 3.1. Characterization of CuO Nanoparticles Dispersion.

2. Fig3. The quality of the figure is inappropriate, I can hardly see something there. Please, improve the figure quality. 

The comment is taken into account and an attempt to improve the quality of the figures was done.

3. Fig. 4. The quality and visibility o the inset are pretty bad. Please, place this inset as a separate sub-figure to make it more visible for readers.

The comment is taken into account and the inset is placed as a separate sub-figure.

4. Table 1. One can see values like this: 1.065 V. Did the authors really get such precision? Probably, it should be much lower. Please, pay attention to it.

The values in Table 1 are calculated by the experimental device. However, the comment of the reviewer is taken into account and the values are rounded for more clarity.

5. Fig. 7. Please, provide here elemental mapping in elemental contrast, it will make the results of investigations much more understandable. 

Generally, the authors agree with the reviewer’s comment that such analysis will make the results more understandable. However, on this stage of our investigations these figures are included only with an aim to demonstrate the scale of corrosion damages on the samples in both corrosive media. More precise investigations concerning the composition of the appeared corrosion products are planned to be realized in the next step.

6. Fig. 10. XPS-analysis - please, clearly point peaks from Cu, O, and Cl in spectra. Right now peaks look a little bit confusing and questionable. 

The comment is taken into consideration and the needed information is added to the figure.

7. I would also like to mention, that the manuscript suffers from a lack of structural analysis. It would be great to provide results of XRD, FTIR, or TEM studies, it will significantly support the authors' conclusions.

As can be seen from the manuscript, TEM investigations have been already realized - Figure 1. The authors are of the opinion that FTIR is not very appropriated method in that case. We agree with the reviewer that XRD investigations are important and we plan to realize them on a later stage with these and other nanoparticles in order to see if their incorporation affect the metallographic structure. However, in that manuscript the main idea was to compare the corrosive behavior of the newly developed coatings in the selected model media and to simile with ordinary zinc.

8. The list of references looks interesting, but I would like to propose to the authors to pay attention to the works [1- 3-. Maybe they will find these works interesting for comparison.

References

  1. Surf. Coat. Technol., 2006, 201, 2621-2632.
  2. Appl. Surf. Sci., 2006, 252, 8043-8049.
  3. Mater. Sci. Eng. C. 2019, 104,109965.

The authors found informative the proposed publications and added them to the list of References. Some comments were also added to our manuscript concerning mainly the antimicrobial effect of Zn – in: Introduction, “Antifouling effect of zinc has also been reported due to antimicrobial and antibacterial properties of zinc-based materials, despite the generally lower toxicity of zinc to aquatic organisms compared to copper [13-15].”

About corrosion protection of steel in different media – in: 3.4. Potentiodynamic Polarization (PDP) Curves and Polarization Resistance. “Different corrosion resistance of hybrid aluminum oxide/titanium nitride coatings produced on stainless steel has earlier been registered in 0.5 M H2SO4, 1 M HCl, and 0.75 M NaCl solutions [29]. The results showed an improvement of the corrosion resistance of the hybrid coatings (compared to pure steel) in most cases, except in NaCl solutions. According to [29], the adhesion of the coating has been considerably affected by the NaCl solution and initiation of crevice and pitting corrosion was observed.”

Reviewer 2 Report

This manuscript presents a comparative study about the corrosion resistance of hybrid zinc coatings in 5% NaCl solution and in artificial sea water. The authors performed many tests and some good results were obtained and inhibition mechanism were discussed.

However, the following are the items that the authors would need to address:

1. The abstract need to improve to clarify the result of the hybrid zinc coatings about anti-corrosion property. The current abstract is mainly composed of tests methods, not appealing to readers.

According to the reviewer’s comment following text was added to the Abstract.

“The tests demonstrate much larger corrosion resistance of the hybrid coating in ASW compared to 5% NaCl solution. The obtained results indicated that the proposed hybrid zinc coating has a potential for antifouling application in marine environment.“

2. The keywords seem like not covering the most important point in the manuscript, such as CuO.

“CuO nanoparticles” was added to the Keywords.

3. In the abstract and conclusion sections, the authors refer to the biofouling property of CuO. But in the whole research there is no biofouling test and results. So, it is suggested to delete the related description about biofouling in abstract and conclusion.

According to the author’s opinion the antimicrobial and antibacterial properties of the zinc-based materials (despite the generally lower toxicity of zinc to aquatic organisms compared to copper), combined with XPS results for the surface corrosion products of the hybrid coating in ASW (presence of Cu) point to such a conclusion. However, the authors agree that biofouling test is necessary to confirm the antifouling properties of the new obtained hybrid zinc coating. For this reason, we noted in the Abstract and Conclusion that “The obtained results indicated that the proposed hybrid zinc coating has a potential for antifouling application in marine environment”.

4. It is suggested to highlight the novelty of the research in the end of the introduction.

 The comment is taken into consideration and the needed text is added at the end of the Introduction part.

5. In line 82, why the suspension was ultrasonicated in ice? Please explained.

The suspension is sonicated to ensure more homogeneous dispersion of CuO powder in water and the procedure is carried out in ice to protect the system from warming and re-agglomeration.

How can you obtain the corrosion current density (Icorr)? Should give the method.

The corrosion current density is calculated from the experimental unit used for recording of the potentiodynamic curves. Тhis can also be done by running tangents to the anode and cathode curves near the corrosion potential and taking into account their intersection point.

  The “2.5. corrosion characterization and CVA studies” section need to improve to clarify the experimental methods, including the PDP scanning rate, the way to get current density and the polarization resistance measurement, et al.

The comment is taken into consideration and the needed text is added.

6. In figure 2, why the ordinate is intensity (%)? Please explained.

This is generally accepted for presenting results of DLS measurements (Volume %, Intensity %, etc.)

7. The authors are suggested to carefully rewrite the conclusion. Similar to the abstract part, lacking of the corrosion resistance effect and the mechanism.

The comment is taken into consideration and the needed corrections are added.
